

# Optimal control of irrupting pest populations in a climate-driven ecosystem

E Penelope Holland[1,*], Rachelle N. Binny[2] and Alex James[3,4,*]

[1] Department of Biology, University of York, York, United Kingdom
[2] Manaaki Whenua Landcare Research, Lincoln, New Zealand
[3] Biomathematics Research Centre, University of Canterbury, Christchurch, New Zealand
[4] Te Pūnaha Matatini, New Zealand
[*] These authors contributed equally to this work.

Corresponding author
Alex James,
alex.james@canterbury.ac.nz

## ABSTRACT

Irruptions of small consumer populations, driven by pulsed resources, can lead to adverse effects including the decline of indigenous species or increased disease spread. Broad-scale pest management to combat such effects benefits from forecasting of irruptions and an assessment of the optimal control conditions for minimising consumer abundance. We use a climate-based consumer-resource model to predict irruptions of a pest species (*Mus musculus*) population in response to masting (episodic synchronous seed production) and extend this model to account for broad-scale pest control of mice using toxic bait. The extended model is used to forecast the magnitude and frequency of pest irruptions under low, moderate and high control levels, and for different timings of control operations. In particular, we assess the optimal control timing required to minimise the frequency with which pests reach 'plague' levels, whilst avoiding excessive toxin use. Model predictions suggest the optimal timing for mouse control in beech forest, with respect to minimising plague time, is mid-September. Of the control regimes considered, a seedfall driven biannual-biennial regime gave the greatest reduction in plague time and plague years for low and moderate control levels. Although inspired by a model validated using house mouse populations in New Zealand forests, our modelling approach is easily adapted for application to other climate-driven systems where broad-scale control is conducted on irrupting pest populations.

# INTRODUCTION

Pulses in food resources can drive irruptions of small consumers and trigger cascading responses in population dynamics across multiple trophic levels (*Ostfeld & Keesing, 2000*). This can result in the decline or extinction of indigenous species (*Innes et al., 2010*) and/or an increase in disease spread (*Jones et al., 1998*), particularly in ecosystems inhabited by pest species. To avoid or mitigate the impact of pest populations on the ecological community, it is necessary to forecast irruptions, often using weather-based forecast models (e.g., *Kelly et al., 2013*; *Magarey & Isard, 2017*), and then optimise the timing and

intensity of management operations to reduce pest abundance at critical times of the year (e.g., *Singleton, Tann & Krebs, 2007*).

The diverse effects of mast seeding - the synchronous intermittent production of large seed crops (*Allen et al., 2012*; *Kelly & Sork, 2002*) - on ecological communities around the world are illustrated well using rodent populations in forests. Yellow-necked mice (*Apodemus flavicollis*) have increased winter survival and rapid population growth following mast seeding in oak (*Quercus robur*) and hornbeam (*Carpinus betulus*) in Biolowieza Primeval Forest in Poland, with a corresponding increase and peak in predator populations such as the pine marten (*Martes martes*) a few months to a year later (*Ostfeld & Keesing, 2000*; *Pucek et al., 1993*). When rodent prey becomes scarce following an irruption, pine martens compensate by increasing consumption of alternative resources including birds and berries (*Jedrzejewska & Jedrzejewski, 1998*). Forest bird populations are also limited by predation by raptors, e.g., buzzards, which undergo prey-switching from rodents to birds following a crash in rodent abundance (*Jedrzejewska & Jedrzejewski, 1998*). In oak (*Quercus* spp.) forest in the eastern U.S., white-footed mice (*Peromyscus leucopus*) respond similarly to acorn masts, with increased winter survival and breeding success (*Jones et al., 1998*). However, in this case mice are also predators, playing an important role in suppressing gypsy moth (*Lymantria dispar*), an invasive and outbreaking species. Defoliation by gypsy moths can delay and reduce mast production, creating a negative feedback loop for the mast-driven mouse populations with catastrophic consequences for tree growth and survival as well as reduced mouse abundance (*Jones et al., 1998*). However, high densities of mice lead to increases in tick (*Ixodes scapularis*) populations, and the associated spread of Lyme disease in humans (*Jones et al., 1998*).

In South America, rodent outbreaks are associated with emerging viral diseases such as hantavirus (*Jaksic & Lima, 2003*). These outbreaks occur after bamboo (*Chusquea* sp. and *Merostachys* sp.) masts, but may also follow heavy rainfall events (*Jaksic & Lima, 2003*), emphasizing the need for ecologically sound pest forecasting models. Introduced bamboo (*Melocanna baccifera*) is a well-used non-timber forest product in India, but rodent (*Rattus* sp. and *Mus* sp.) migration towards agricultural areas following mast events can have devastating effects on standing crops and stored grains, as well as increasing the risk of infection for rodent-borne diseases (*Biswas, Kumar & Mittal, 2016*). It has also been suggested that masting by introduced Asian bamboos into North America, particularly the Pacific Northwest, poses a health risk to humans as a result of population irruptions and subsequent dispersal of deer mice (*Peromyscus maniculatus*, a hantavirus carrier) following mast seed depletion (*Smith, Gomulkiewicz & Mack, 2015*).

Broad-scale management of mouse populations to reduce damaging effects is typically attempted through the application of baits laced with rodenticide toxin, applied either via aerial drops or ground-based bait stations (*Singleton et al., 2010*). In Australia, for example, plaguing mouse populations can exceed densities of 1,000 mice per hectare, causing significant damage to cereal crops (*Singleton et al., 2001*). Sterilised wheat grains coated with zinc phosphide toxin are applied in and around crops, aerially or using calibrated fertiliser spreaders, and can reduce the mouse population by 40–98% (*Mutze & Sinclair, 2004*).

In New Zealand, invasive house mouse (*Mus musculus)* populations undergo episodic outbreaks in response to high seedfall, particularly in the heavy masts of native beech forest (*Nothofagus* species) (*Fitzgerald et al., 1996*; *King, 1983*; *Ruscoe, 2001*) or rimu (*Dacrydium cupressinum*) (*Ruscoe et al., 2004*). Irruptions in beech forest occur predominantly in late summer to autumn (i.e., February–May) (*Ruscoe et al., 2003*; *Wardle, 1984*) and are accompanied by an increase in abundance of other seed consumers, including ship rats (*Rattus rattus*) (*Studholme, 2000*) and kiore (*R. exulans)* (*Ruscoe & Pech, 2010*). Small mammal predators, predominantly invasive stoats (*Mustela erminea*), benefit from the irruptions of prey and increase after a delay caused by seasonal breeding (*King, 1983*). These dynamics are harmful to a wide range of native fauna. For instance, in addition to fallen seed, mouse and rat diets comprise invertebrates (*Fitzgerald et al., 1996*; *Jones & Toft, 2006*; *Miller & Miller, 1995*; *Ruscoe & Murphy, 2005*), bird chicks and eggs (*O'Donnell & Phillipson, 1996*; *Wilson et al., 2006*; *Wilson et al., 1998*), and lizards (*Norbury et al., 2014*). Stoats also have a flexible diet, and in masting forests will switch from mice as their primary food source, to predating on birds and invertebrates when mice become scarce (*Murphy et al., 2016*; *Wilson et al., 1998*).

Despite the threat posed to native biota, there is currently no broad-scale control targeting mice alone in mainland New Zealand. This is primarily due to higher prioritisation of control for other small mammal pest targets, e.g., common brushtail possum (*Trichosurus vulpecula*), rats (*Rattus* sp.) and mustelids (*Mustela* sp.), which are considered to pose a greater risk to native flora and fauna (*Innes et al., 2010*; *Ruscoe & Pech, 2010*). In addition, the lack of cost-effective mouse-specific control tools and public concern around widespread toxin use, means that the broad-scale management of mice on mainland New Zealand still presents a major challenge (*Ruscoe & Pech, 2010*). Ground-based control (e.g., trapping) targeting mice alone has been undertaken on smaller spatial scales within predator-fenced sanctuaries, where all other vertebrate pests have been eradicated, and has been shown to confer benefits to biodiversity (*Watts et al., 2017*). Broad-scale multi-species control operations involving aerially applied bait laced within sodium fluoroacetate (1,080) toxin are undertaken in New Zealand to control rodents, possums and stoats (via secondary poisoning when stoats consume poisoned rodents (*Murphy et al., 1999*)). These operations have been partially successful in reducing mouse abundance but less so than for other small mammals (*Elliott & Kemp, 2016*), likely due to lower uptake of the 1,080 bait by mice compared to other targets (*Fisher & Airey, 2009*; *O'Connor, Morriss & Murphy, 2005*). There has been some success with mouse eradication programmes on New Zealand's offshore islands, predominantly through aerial application of the second-generation anti-coagulant toxin brodifacoum (*Mackay, Russell & Murphy, 2007*). However, long-term broad-scale brodifacoum use is avoided on mainland New Zealand due to its persistence in the environment and risks to non-target species (*Eason et al., 2002*).

Control targeting only a subset of predators in an ecosystem may lead to an increase in abundance of other smaller predators, an effect termed 'mesopredator release', causing a subsequent decline in indigenous prey species (*Crooks & Soulé, 1999*; *Ritchie & Johnson, 2009*). In New Zealand, mesopredator release of mouse populations has been observed both on islands (*Simberloff, 2002*) and on the mainland (*Norbury et al., 2013*). It is therefore

becoming increasingly important to have the understanding and technologies in place to effectively forecast and manage irrupting mouse populations over large spatial scales. In particular, the optimal timing for broad-scale mouse control on mainland New Zealand remains a critical knowledge gap. For eradication of mice on islands, the preferred season for control is winter to early spring when food is likely to be limiting because this maximizes bait up-take (*Broome et al., 2017*). The optimal timing for broad-scale aerial 1,080 control targeting irrupting rodents and stoats is between July and November in a mast year, determined with the aim of minimising rat abundance. However, operational logistics (e.g., availability of helicopters), weather and legal requirements also place significant constraints on timing (*Elliott & Kemp, 2016*). *Innes et al. (1995)* proposed that broad-scale aerial operations targeting ship rats (*Rattus rattus*) to protect nesting birds should coincide with the onset of nesting each year. The conceptual model of *Wilson et al. (1998)* for mouse dynamics in beech forest suggests conducting control in November prior to a mast would have little effect, while the November after a mast would be too late (see e.g., Fig. 5 in *Wilson et al. (1998)*). However, conducting control in February at the start of a mast and/or May during heavy seedfall may be optimal.

Models for irrupting mouse populations require good predictions of the size and timing of masts. Temperature and rainfall in the years before the mast event are almost always the primary cue for mast seeding (*Janzen, 1971*). *Kelly et al. (2013)* proposed a generic and widely applicable model using the change in temperature in a set period over the previous two years ($\Delta T$) as the sole predictor. This model offered much improved predictions over other simple models and has been shown to be applicable to a wide range of plant species around the world, including oak and many New Zealand species, either as a cue or a proximate driver of masting (*Kelly et al., 2013*; *Pearse, Koenig & Knops, 2014*).

*Holland et al. (2015)* previously developed a climate-based consumer-resource model for mouse irruptions in masting forests, parameterised using long-term temperature, hard beech (*Fuscospora truncata*) seedfall and house mouse (*Mus musculus*) abundance data from mixed beech-podocarp-broadleaved forest in Orongorongo Valley (OV), New Zealand. Seedfall was predicted using the $\Delta T$ model (*Kelly et al., 2013*). In this paper, we extend the *Holland et al. (2015)* model of mouse population dynamics driven by pulses in food resource, to account for broad-scale mouse control. We use the extended model to forecast population irruptions (timing and size) and the impact of pest control on mouse populations, to assess if and how we can avoid 'plague' levels of mice while also avoiding excessive poison use. Predictions of the impacts of tailored vs. untailored vs. no control will be crucial for effective and efficient broad-scale management of irrupting mouse populations.

## METHODS

### Consumer–resource model

The underlying consumer-resource model is the best-fit model developed by *Holland et al. (2015)*. Relative mouse abundance $M(t)$ is quantified by an index: captures per 100 trap nights (C/100TN) (*Ruscoe, Goldsmith & Choquenot, 2001*). The rate of change of $M$ over

time $t$ (years) is given by

$$\frac{dM}{dt} = \left(\alpha g(F) - \mu_1 - \mu_2 M - B(t)\right)M, \tag{1}$$

Food availability $F$ (seeds m$^{-2}$) is predicted by the functional response $g(F)$ and $\alpha$ is the demographic efficiency of mice (i.e., efficiency at converting food into recruitment for the mouse population). The total mortality rate is $\mu_1 - \mu_2 M - B(t)$, where the parameters $\mu_1$ and $\mu_2$ are density-independent and density-dependent mortality rates respectively. They may both be positive or negative depending on non-food related processes, e.g., predation, social interactions, Allee effects. In this paper, we extend the previously published model by adding $B(t)$, which is a time-dependent, density-independent rate of mortality due to control by bait application (see below).

The original model tested four candidate models for the food availability functional response. The best fit was a Holling II (Ivlev) function where $c$ (seeds m$^{-2}$ mouse$^{-1}$ year$^{-1}$) is the maximum per capita feeding rate and $e$ ((seeds m$^{-2}$)$^{-1}$) is a measure of foraging efficiency:

$$g(F) = c(1 - \exp(-eF)). \tag{2}$$

The rate of change of available food over time is modelled by

$$\frac{dF}{dt} = S(t) - hF - g(F)M, \tag{3}$$

where the second term, $hF$, describes the change in available food that happens throughout the year at a constant rate $h$ (year$^{-1}$) unrelated to mouse density and the third term, $g(F)M$, describes the rate of seed consumption by mice. The first term, $S(t)$, describes the rate at which food is delivered as a function of time, with

$$S(t) = \begin{cases} \dfrac{F_y}{0.25} & \text{if } 0 \leq t - \text{floor}(t) < 0.25 \\ 0 & \text{otherwise.} \end{cases} \tag{4}$$

The floor function rounds $t$ down to the largest integer smaller than $t$. Thus, during the $y$th annual cycle, a total amount of seed $F_y$ is produced, which is delivered at a constant rate over the first quarter of the year (nominally February–April). At the start of each year we set $F(t) = 0$, i.e., seed is not carried over between years. The annual seedfall amount $F_y$, was determined by a climate induced seedfall model (see below). All parameter values are given in Table 1 and were those determined as best-fit parameter values by *Holland et al. (2015)* (Table 2 in *Holland et al. (2015)*). These were chosen by modelling mouse density over 25 years using observed annual seedfall data from the OV (starting February 1972) as the annual values of seed $F_y$. Mouse density at the start of each quarter was extracted from the continuous-time model prediction. These predicted values were compared to observed quarterly mouse density data from the OV collected over the same time period (quarterly, February 1972 –November 1996) and best-fit parameter estimates chosen by minimising the root mean square error. *Holland et al. (2015)* showed that with these best-fit parameter values the model predicted all major outbreaks in mouse density occurring in the 25 year
**Table 1  Parameters and variables.**

|  | Symbol | Value | Units |
|---|---|---|---|
| **Parameters** | | | |
| Demographic efficiency | $\alpha$ | 1 | mice (seeds m$^{-2}$)$^{-1}$ |
| Density independent birth | $\mu_1$ | $-1.23$ | year$^{-1}$ |
| Density dependent mortality | $\mu_2$ | 0.76 | mouse$^{-1}$ year$^{-1}$ |
| Maximum per capita feeding rate | $c$ | 6.74 | seeds m$^{-2}$ mouse$^{-1}$ year$^{-1}$ |
| Foraging efficiency | $e$ | 1.08 | (seeds m$^{-2}$)$^{-1}$ |
| Seedfall decay | $h$ | 9.48 | year$^{-1}$ |
| Bait decay | $d$ | 50 | year$^{-1}$ |
| Control level | $B_0$ | 50, 100, 150 | – |
| **Variables** | | | |
| Mouse density | $M(t)$ | – | mice (measured as an index = captures per 100 trap nights (C/100TN)) |
| Available resource density | $F(t)$ | – | seeds m$^{-2}$ |
| Resource delivery rate | $S(t)$ | – | seeds m$^{-2}$ year$^{-1}$ |
| Mortality rate due to control | $B(t)$ | – | year$^{-1}$ |

observed mouse density (C/100TN) time series, although it tended to slightly under-predict the magnitude of outbreaks.

During a control year, bait is applied as an impulse function at a specific time point $t_i^*$, such that $t - t_i^*$ describes the time since the $i$th bait application. After application, bait degrades according to a simple decay function $B_0 \exp(-d(t - t_i^*))$, where $d$ is the decay rate. The constant $B_0$ governs peak bait availability, i.e., at the time of application $(t - t_i^* = 0)$. Therefore, bait availability $b_i(t)$ from the $i$th application is described by a piecewise function

$$b_i(t) = \begin{cases} B_0 \exp\left(-d\left(t - t_i^*\right)\right), \text{if } t_i^* \leq t < t_{i+1}^* \\ 0, \quad \text{otherwise,} \end{cases} \tag{5}$$

and overall bait availability $B(t)$ at time $t$ is given by

$$B(t) = \sum_{i=1}^{n-1} b_i(t), \tag{6}$$

for $n$ bait applications. In the absence of mice, bait is considered to be inactive after one month, so we choose $d = 50$ (meaning that $b_i(t)$ has decayed to <2% of its original size one month after the $i$th application, and <0.02% of its original size after two months. It is presumed that, compared to this decay rate, the effect of mouse predation on bait levels is negligible. Note that the actual value of $B_0$ is defined later in terms of the kill success rate.

## Climate-induced seedfall model

A 1,000 year normally distributed temperature time series $T_1, T_2, \ldots, T_{1,000}$, was generated where

$$T_y \sim N(14, 1). \tag{7}$$

This represents historical mean summer temperatures (daily average for the three month period January to March) in the Orongorongo Valley (hereafter OV), 1972–2014. Randomly generating time-series for the OV in this way was shown to be a valid approach by *Holland & James (2015)*. From this, a time series was calculated using the $\Delta$T model of *Kelly et al. (2013)*, where

$$\Delta T_y = T_y - T_{y-1}. \tag{8}$$

Annual seedfall predictions were made using the following linear relationship fitted to observed OV beech seedfall data (1972–1996) by *Holland et al. (2015)*:

$$\log_{10} F_y = 0.33 + 0.97 \Delta T_y + \epsilon_y. \tag{9}$$

The noise term was chosen to have distribution $\epsilon_y \sim N(0, 1.3)$ to give a correlation between change in temperature and seedfall of $r^2 \approx 0.54$ corresponding to the findings of *Kelly et al. (2013)*. These seedfall time series were used as annual inputs of $F_y$ to the mouse model above. Mouse density (C/100TN) was simulated for each of the control scenarios above, with $M(0) = 1.0$.

## Plague definitions

We define the mouse density plague level, $M_p = 2.02$, to be the maximum mouse density in a median seedfall year if the initial mouse density is one, i.e., $M(0) = 1$, (see Fig. 1). Using this definition, if the mouse density was 1 at the start of the year ($M(0) = 1$) 50% of years would be defined as *plague years* i.e., the mouse density 'just' reached plague level at some point during the year. In a longer time series where no control measures are applied (Fig. 2) and the density is continuous across years, i.e., mouse density at the start of the year is the density at the end of the previous year, the plague level definition is not changed and the proportion of years that are plague years is much higher, 85%. Specific thresholds for what constitutes a mouse 'plague' or eruption in New Zealand have not been formally defined in the literature; it is difficult to measure exact population densities, and it is not known exactly at what threshold level mice may have an impact on native biodiversity. We therefore use the term 'plague', to mean greater than average population abundance for an extended period of time, i.e., demonstrably not an undetectable population and therefore likely to have some impact, and a convenient reference point with which to compare scenarios. In the 25 year time-series of observed mouse abundance from the OV, when mouse abundance was above 2.02 the population tended to be undergoing one of seven larger eruptions. In addition, mouse abundance was above the 2.02 level in 80% of years, suggesting that our definition for plague level and estimate of proportion of plague years are reasonable in this context. We also define *plague time,* the proportion of time during which the mouse density is above the plague level. For example, in Fig. 1 the plague time for the mast year trajectory is 0.89. In the long term time series of Fig. 2 the plague time is 0.71. Finally we define the *plague size*, the highest mouse density during the plague period. In the Fig. 1 example, the plague size for the mast trajectory is 6.93. In the long term series (Fig. 2), the expected plague size, given that a plague occurs, is 5.7. Higher values of $M_p$ could be used with the same plague definitions given here, with qualitatively similar results.

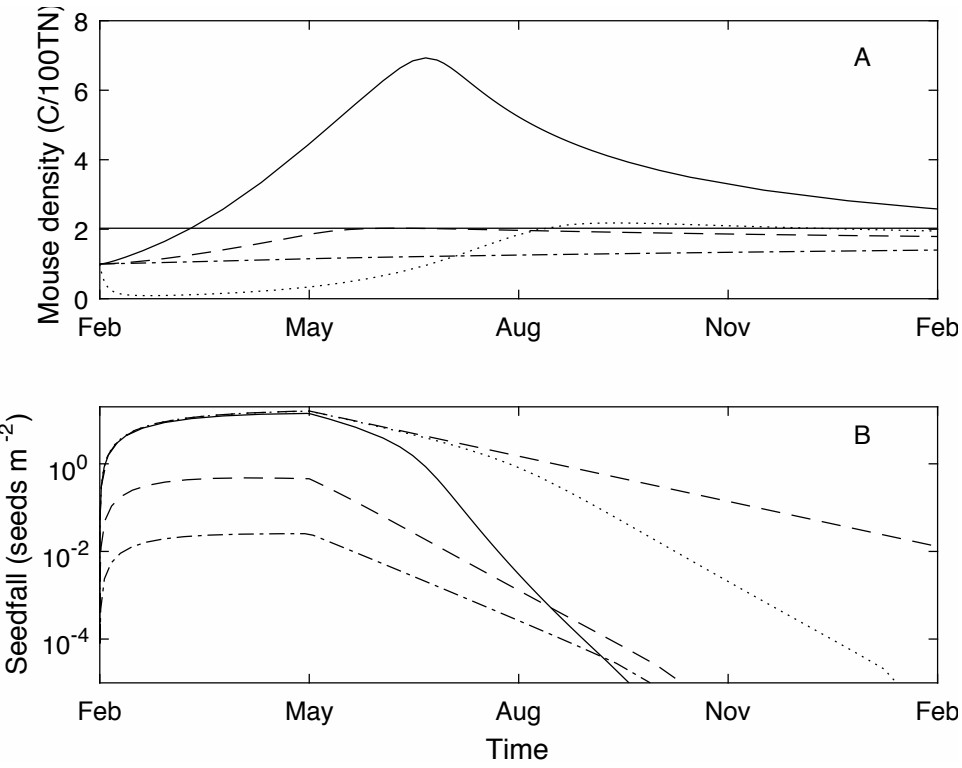

**Figure 1  Example one year (Feb-Feb) time series of mouse density and seedfall.** Example one year (Feb-Feb) time series of mouse density (C/100TN) (A) and seedfall (seeds m$^{-2}$) (B, log scale) through time, modelled using three different seedfall levels: high (a mast year, the 75th percentile of the seedfall distribution, black solid line); median (50th percentile of the seedfall distribution, blue dashed line); low (25th percentile of the seedfall distribution, green dot-dashed line). The mouse plague level is defined such that, for an initial mouse density of 1, if the seedfall is at or below the median level, there is no plague (grey horizontal line, A only). Note that the start of the year coincides with the start of the seedfall season. When high level control takes place at the start of the seedfall season (February 1) in a mast year (black dotted line, A and B), the mouse density is much reduced compared to no control in a mast year (c.f. black solid line, A) and seed remains available until the end of the year (c.f. black solid line, B). With control, the seedfall is closer to the seedfall in the absence of mice (red dashed line, B only).

## Control definitions

The strength of the control impulse is governed by the parameter $B_0$, which is the value of $B(t)$ at the time of bait application $t = t_i^*$ (Eqs. (5)–(6)). The absolute value of the control impulse is of little practical use in modelling terms, though operationally it relates to control effort and a parameter value could be calibrated for a given operation. A more appropriate measure of control size is the *control success*, typically assessed in terms of percentage kill, defined here as the relative decrease in the mouse density one month after the control impulse. The control success will change depending on the current mast level and mouse density but percentage kill is still a useful and widely used measure (*Elliott & Kemp, 2016*; *Innes et al., 1995*). In the example of Fig. 1 (black dotted line), where $B_0 = 150$ and control is applied at the start of the mast season, control success is 88%. Even with this relatively high level of control (consistent with historic broad-scale aerial poison operations

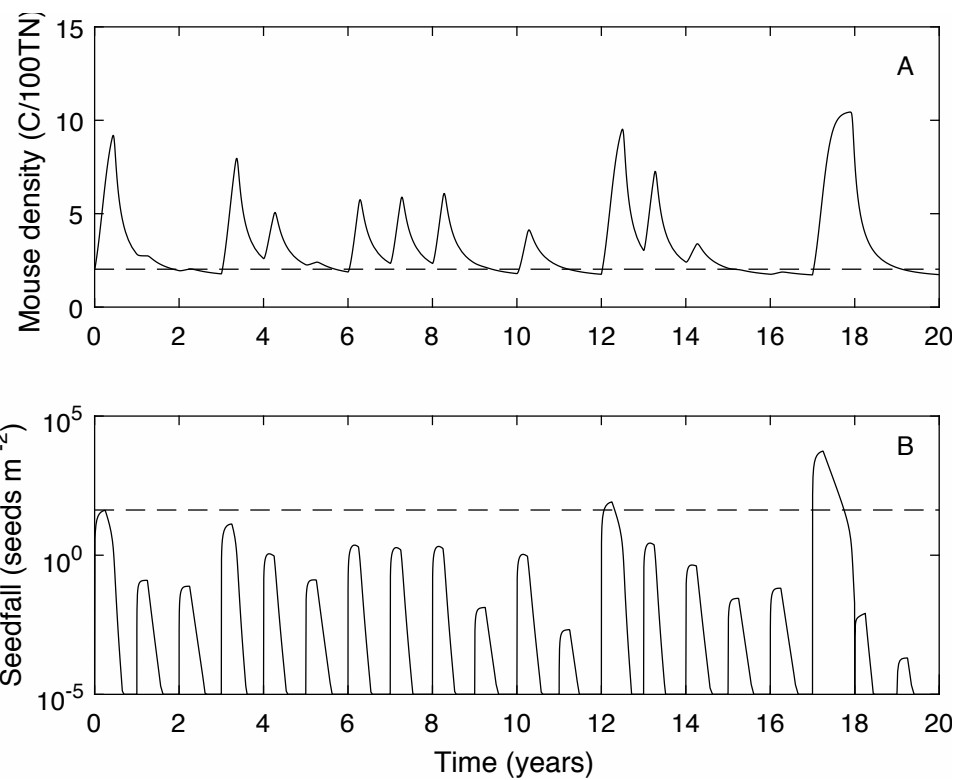

**Figure 2** **A 20 year example time series for mouse density and seedfall in the absence of control.** A 20 year example time series for mouse density (A) and seedfall (B, log scale) in the absence of control. The dashed lines show the defined plague level for the mouse population and the seedfall mast level. The plague threshold is exceeded in 85% of years and for 71% of the total time.

for rodents (*Innes et al., 1995*)), the mouse density at the end of the year remains high as the population recovers, and the mouse density rises above the plague level during September. However, both the plague time (0.31) and the plague size (2.18) are much lower than in the uncontrolled mast year. Crucially, for the success of native species that rely on beech mast for food, the seedfall level is much higher in the controlled scenario compared to the uncontrolled.

## Scenarios

To obtain precise estimates for expected plague time, plague size and proportion of plague years under different control scenarios, we ran the extended model (Eqs. (1)–(6)) or a 1,000 year long simulation with $M(0) = 1$, using a seedfall time series generated with (Eqs. (8)–(9)) from a temperature time series generated using Eq. (7). In this longer simulation, the initial mouse density is not reset to 1 each year but continues with the value at the end of the previous year. Seedfall is reset to zero at the beginning of each year to reflect the inedibility of the previous year's seed. Note that all the control scenarios used the same underlying seedfall series.

## Optimal control timing

Initially, we assume that control can only be applied once each year and we examine three levels of control: low, medium and high, $B_0 = 50, 100, 150$, respectively. For each level of control, we used the 1,000 year weather/seedfall time series, applying control at a range of times throughout the year. We calculated the proportion of plague years, proportion of time above plague level (i.e., *plague time*), and average plague size for each control application time.

## Tailored control

We now consider how other control options with similar costs could offer greater benefit if tailored. An alternative to annual control could be a biannual-biennial regime, i.e., control is applied twice a year every two years. In some cases this may be more cost effective as resources only need to be acquired every other year; it may also be socially advantageous if the application of poison is somewhat controversial and its use needs to be limited. As operational costs may be considerable (particularly labour and transport/flight-time), and the amount of bait applied contributes relatively little to the overall cost of each control dose, we do not consider control via very frequent smaller doses here. We used the 1,000 year seedfall simulation to compare three biannual-biennial regimes with no control and the annual control regime described above:

1. Regular biennial control: Control occurs every second year and is executed in early September and a month later in early October, i.e., straddling the optimal control period (see 'Results').
2. Seedfall determined control: Control occurs in years when seedfall is above the median (i.e., with the same long term average frequency as regular biennial control), in early September and early October.
3. Climate determined control: Control occurs in years when the seedfall temperature driver $\Delta T$ is above the median (i.e., with the same long term average frequency as regular biennial control but with more opportunity for error in true/false seedfall prediction), in early September and early October.

   Regime two could be used if the seedfall could be measured early enough to plan a control application that year. In cases where this was too late to muster a control application then regime three may be of use.

# RESULTS

When mice are not controlled, 85% of years are plague years, the mouse density is above the plague level for 71% of the time and the average plague size is 5.71 (see Fig. 2 for a 20 year time series example).

## Optimal control timing

As the control time is changed across the year the plague time varies (Fig. 3A). The different control levels (low, medium and high) have a much stronger effect on the plague time than the control timing. For example, under the low control regime, where the control success rate is between 50 and 60%, the optimal control timing to reduce plague time is spring

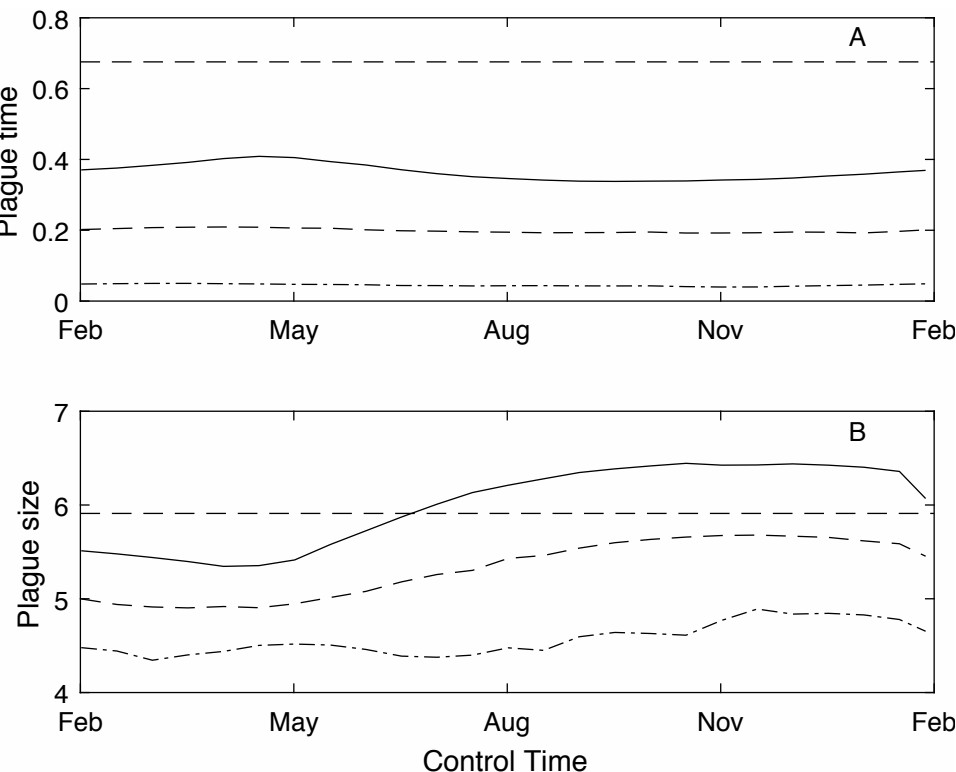

**Figure 3** **The effect of timing of annual control on plague time and plague size.** The effect of timing of annual control on plague time (A) and plague size (B), for low (red), medium (blue dashed) and high (green dot-dashed) control levels, and compared to no control (black dashed). If control has a low success rate then optimal control timing is mid-September. However, if control is more effective then there is little difference in plague time if control is applied at different times throughout the year. Plague size is more strongly affected by control timing, particularly for low and medium control levels.

(mid-September) before the mast season starts when plague time is 0.36. Conversely the lowest reduction is seen in autumn (late April) at the end of the mast season which reduces the plague time to 0.42. The optimal control time at medium and high control levels is also around mid-September. Medium control, which has an 80–85% success rate, reduces the plague time to 0.20, while high control has a 92–94% success rate and reduces the plague time to 0.045.

In contrast, the expected plague size (Fig. 3B) appears to be strongly affected by control timing, in particular for low and medium control levels. The optimal timing to reduce plague time gives the least reduction in expected plague size. Initially this seems counter-intuitive but less so after recalling the definition of expected plague size (the maximum mouse density *given* that there is a plague). On examination of the time series (Fig. 4) for control during the mast season (March—Figs. 4A, 4C, 4E) and after (September—Figs. 4B, 4D, 4F) we see that the small reduction in plague time between the control timings comes from the years where the plague threshold is only just exceeded. For low control effort both timings have relatively little impact on the maximum mouse density reached for large plagues, but in post mast control (September) the timing is appropriate to reduce
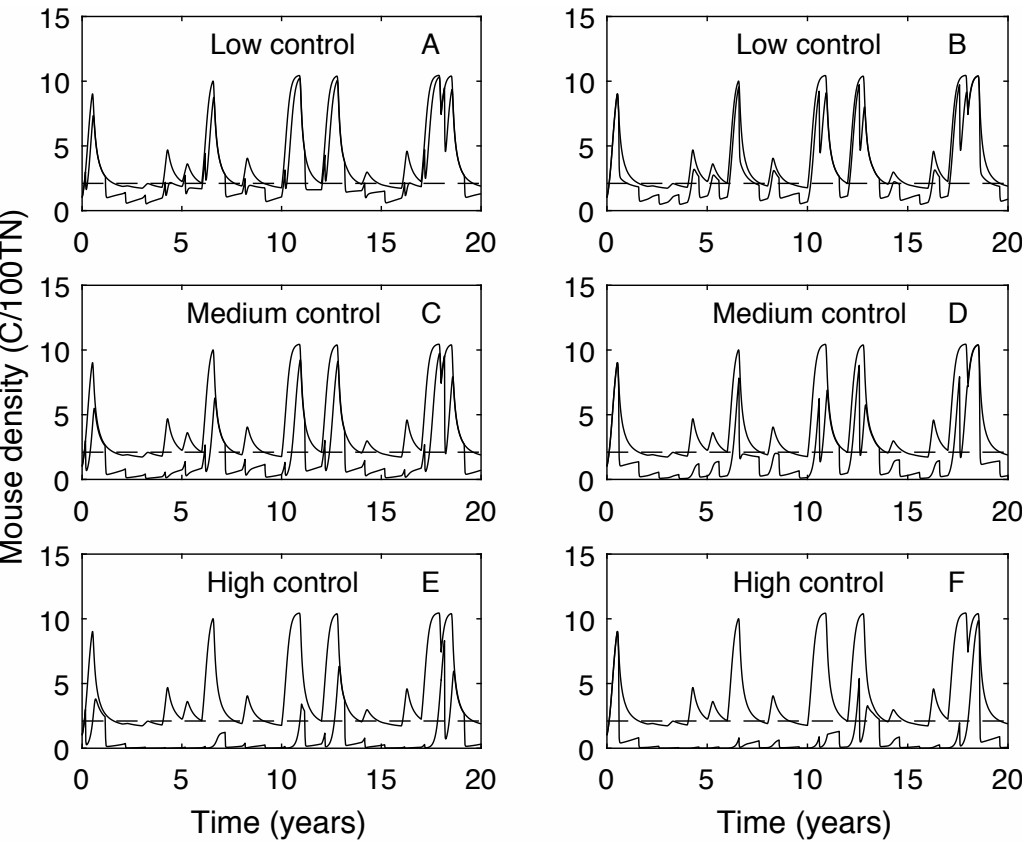

**Figure 4** **Time series of mouse density for each of the three control levels applied annually, compared against mouse density with no control.** Time series of mouse density (C/100TN) for each of the three control levels (low, medium and high) applied annually (red lines), compared against mouse density with no control (black lines). The black dashed line shows the mouse density plague level. A, C, E show control applied in autumn (March), B, D, F show control applied in early spring (September). Low control stops plagues in only the years with the lowest plague size. Contrastingly, high control reduces mouse density to below the plague threshold in almost all years but in the highest plague years mouse populations still persist.

mouse density to below the plague threshold for smaller plagues. This means that, for low level control post mast season small plagues are often avoided but if control is during the mast the small plagues still occur. Larger plagues still occur and reach similar maximum densities as in the absence of control for both timings. When calculating an expected plague size over the entire time-series, smaller plagues will reduce the average. Therefore, while the larger expected plague size for low control may seem counter-intuitive, the reduction in small plagues driving this is actually a desirable outcome. Using an alternative metric of plague severity, for example the expected density over the entire time-series, loses this subtlety and gives results similar to the plague time metric.

### Tailored control
Seedfall- and climate-determined biennial-biannual control regimes are more effective in reducing proportion of plague years and time above plague levels than a regular

**Table 2 Effectiveness of control regimes.** Plague time, proportion of time density is above the plague level. Plague years, proportion of years maximum density exceeds the plague level. Plague size (C/100TN) maximum density during plague years. All values calculated from 1,000 year time series. Control is carried out in mid-September for annual control or two weeks either side for regular biennial control. For each control level, the minimum expected plague time, plague years and plague size across the different regimes are shown in bold text.

| Level Regime | Plague time | | | Plague years | | | Plague size (C/100TN) | | |
|---|---|---|---|---|---|---|---|---|---|
| | Low | Med | High | Low | Med | High | Low | Med | High |
| None | 0.72 | | | 0.85 | | | 5.78 | | |
| Annual | 0.35 | 0.20 | **0.042** | 0.56 | 0.43 | 0.15 | 6.39 | 5.53 | **4.39** |
| Biannual | | | | | | | | | |
| Regular biennial | 0.42 | 0.23 | 0.056 | 0.62 | 0.45 | 0.16 | **5.96** | **5.48** | 4.73 |
| Seedfall driven | **0.34** | **0.18** | 0.048 | **0.55** | **0.41** | 0.15 | 6.43 | 5.96 | 5.16 |
| Climate driven | 0.39 | 0.20 | 0.057 | 0.57 | **0.41** | **0.13** | 6.27 | 5.93 | 5.41 |

biennial-biannual control regime (Table 2). For both the low and medium control levels a seedfall driven biennial-biannual control regime is more effective than annual control (Table 2). If the control trigger is based on climatic variables rather than actual seedfall, the biennial-biannual regime is slightly less effective than annual control, though the differences in plague time between these methods is relatively small.

# DISCUSSION

Management to mitigate the adverse effects of irrupting small consumers should be optimised to ensure that the limited resources available to managers are implemented to have the greatest impact and to meet control objectives. In this work, we offer insights into the dynamics of an irrupting rodent population undergoing broad-scale aerial control in an ecosystem with pulsed resources. It is clear from our results that the timing and frequency of control can affect the time spent above the plague threshold. While we have focused on mouse irruptions in beech forest, insights from this work have clear ramifications for dynamics of predators, prey and disease spread in other systems with climate-driven pulsed resources and outbreaking consumers worldwide.

There is currently no broad-scale control of mice alone in mainland New Zealand, due, in part, to the higher prioritisation of other vertebrate pest targets. In addition, achieving a large reduction in mouse abundance at large spatial scale is difficult with the control tools currently available on the mainland (*Ruscoe & Pech, 2010*). As a result, considerably less is known about the optimal control practices when targeting mice, compared to other small mammals. In a year of high seedfall and in the absence of control, our model predicts that mouse populations exceed the plague threshold and drive a rapid decline in the amount of available seed. As seedfall is depleted, mouse density decreases but still remains above the plague threshold by the end of the mast year. As well as the local impacts of this prolonged high mouse density, there is a risk that after seedfall depletion mice may disperse out of beech forest into other adjacent areas or habitats (*Choquenot & Ruscoe, 2000*; *Ostfeld & Keesing, 2000*). Dispersal of outbreaking rodents has been reported globally
and contributes to disease spread (*Deter et al., 2008*; *Smith, Gomulkiewicz & Mack, 2015*), damage to agriculture (*Biswas, Kumar & Mittal, 2016*; *Newsome, 1969*; *Ruscoe, 1996*) or declines in indigenous biota (*Smith, Dickman & Banks, 2016*). However, conducting high level mouse control in mast years can successfully reduce mouse density to below plague level and to an extent where seed abundance is closer to that observed in the absence of mice. This should provide more food resource for other indigenous consumers competing for seedfall, and reduce the risks of dispersal and predation by mice and stoats.

Our model quantifies relative mouse abundance as an index of captures per 100 trap nights. Measuring actual mouse density in the field is difficult and costly, therefore indices such as C/100TN or a rodent tracking index (i.e., mean percentage of 'run-through' tracking tunnels containing mouse prints per line) are commonly used (*Blackwell, Potter & McLennan, 2002*; *Ruscoe, Goldsmith & Choquenot, 2001*). Quantifying relative abundance in this way facilitates model parameterisation and validation of model predictions using observation data that is more readily available to managers. Our model assumes logistic-type density dependence in the mouse population, which has been shown to be a good description of density-dependent dynamics in small rodent populations (see e.g., *Turchin & Ostfeld (1997)*). The best-fit parameters were similar to those used in other models of house mouse population dynamics, for example the model of *Choquenot & Ruscoe (2000)* also had a positive density-independent growth term and a negative density-dependent growth term.

The three values chosen for peak bait availability to model low, moderate and high control levels, gave control success rates (measured as percentage kill) in the ranges 50–60%, 80–85% and 92–94%, respectively. How these ranges relate to success for real control operations, that aim to suppress as opposed to eradicate mouse populations, will depend on control objectives. To date, very little has been published on the density-impact relationships for mouse abundance and biodiversity in New Zealand. These knowledge deficits currently present a major barrier for managers, both in terms of setting conservation aims and measuring success of mouse control operations. However, our modelling framework provides predictions of mouse abundance and kill rates that can be readily applied to real control operations, as research advances in this area and new thresholds for successful suppression of mouse populations on mainland New Zealand are set.

Our model predicts that the optimal timing for mouse control in beech forest, with respect to minimising plague time, is mid-September. This timing fits within the recommended range for broad-scale aerial 1,080 control targeting rats and stoats (*Elliott & Kemp, 2016*). Across all the regimes considered here, the seedfall driven biannual-biennial regime gave the greatest reduction in plague time and plague years for low and moderate control levels, although the differences between regimes were relatively small. Managers will need to take additional factors into consideration when selecting an optimal approach, for instance the benefits of seedfall driven control need to be weighed against the cost and effort associated with collecting the necessary seedfall data, while temperature data required for the climate driven regime is readily available.

An important advantage of this modelling approach is its simplicity and generality. This work considered a case study of optimal control for mouse populations in New Zealand

hard beech forest, where the aim is to minimise plague time. Different compositions of masting species in other forests will drive slightly different seedfall and mouse dynamics. For example, *Ruscoe et al. (2004)* reported a later onset of mouse population increase due to heavy rimu (*D. cupressium*) masting occurring two to three months later than in beech forest. Therefore, optimal control conditions will likely differ for other forest compositions. Nonetheless, our model and approach could be easily adapted for application to other habitats with climate-driven pulsed resources, for which temperature, resource, and consumer abundance data is available for parameterisation. Similarly, it would be straightforward to alter our model to account for additional drivers of population irruptions, such as the effect of rainfall alongside bamboo masts on rodent outbreaks in South America (*Jaksic & Lima, 2003*), or to adjust the thresholds for conducting control.

Interactions between mice and their competitors or predators are captured implicitly in the model via the density-independent and density-dependent growth rates. However, making these interactions explicit by including rate of change of equations for other interacting species in the system, could offer additional insights, e.g., into cascading responses across different trophic levels. For instance, an equation for stoat density could be included in the model to consider the effectiveness of targeting mice as vectors for secondary poisoning of stoats. In addition, this work could be extended to relate the mouse densities expected under different control regimes to outcomes for indigenous biota, as this will be another key factor for determining the optimal approach and assessing whether conservation objectives are being met.

In this work, we have attempted to maintain approximately equal costs across each control scenario by comparing regimes with similar long-term frequency (e.g., annual vs. biennual-biannual). We assume that low, moderate and high effort poisoning will likely have similar overall costs since the operational costs (e.g., aerial transport and/or application, labour costs) are relatively large and constant compared to equipment costs (e.g., traps, bait). However, a cost-benefit analysis of higher and lower frequency control regimes could also be undertaken.

Our results assume the same distribution of summer temperatures over a 1,000 year time-series; however the effects of climate change could be investigated by relaxing this assumption. If consumer-resource dynamics are altered due to climate change, our model could be useful for guiding how management timing and intensity should be modified to still be effective in reducing the mouse population. For example, our model would be compatible with a recently developed framework, based on the *Kelly et al. (2013)* model, that uses climate projections to assess whether climate change might affect the frequency or spatial extent of beech forest masts (*Barron et al., 2016*).

## CONCLUSIONS

With large-scale predator control campaigns causing pest control to ramp up across New Zealand's mainland (*Russell et al., 2015*), there are opportunities to answer increasingly complex questions around the impacts of broad-scale invasive pest management for ecosystems, and to determine optimal control practices. We have attempted to fill an

important knowledge gap concerning broad-scale control of irrupting mouse populations in masting beech forest, however a deeper understanding of climate-driven consumer-resource dynamics and control outcomes will benefit managers globally. In a rapidly changing world, having the modelling tools in place to make good predictions about the behaviour of such systems, puts us in a stronger position to anticipate and mitigate the potential adverse effects of change.

## ACKNOWLEDGEMENTS

The authors thank John Innes for helpful discussions, and Andrea Byrom and Roger Pech for useful comments on manuscript drafts.

### Funding
The authors received no funding for this work.

### Competing Interests
The authors declare there are no competing interests.

### Author Contributions
- E Penelope Holland and Rachelle N. Binny analyzed the data, prepared figures and/or tables, authored or reviewed drafts of the paper, approved the final draft.
- Alex James analyzed the data, prepared figures and/or tables, approved the final draft.

### Data Availability
Data used to parameterise the model (temperature, seedfall, and trap catch of mice) are the same as for *Holland et al. (2015)* and are publicly available from the Landcare Research Manaaki Whenua DataStore database at the URL https://datastore.landcareresearch.co.nz/dataset/climate-driven-consumer-resource-models-data or at the DOI 10.7931/J2W66HPB.

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
