# Peer review of "Optimal control of irrupting pest populations in a climate-driven ecosystem"

_PeerJ, doi:10.7717/peerj.6146_

## Round 0.1 · original submission · Minor Revisions

I commend the authors on an exceptionally clear and well-written manuscript. Both reviewers found the approach valid and results convincing and suggested only very minor corrections.

I apologize for the delay in completing my review. The manuscript arrived at the same time as several others and at a busy time for me personally, resulting in a substantial backlog.

·

Basic reporting

In this manuscript, Holland and colleagues expand a previous model of consumer population dynamics with pulsed resources the authors investigated in 2015 (Holland et al. 2015) in order to study the optimal control of irrupting pest population (case study
being the broad-scale control of house mice outbreaks in New Zealand).

The authors convincingly show that the control level (*i.e.* the number of baits) impacts both the plague size (*i.e.* the "highest mouse density during the plague period") and the plague time (*i.e.* the "proportion of time during which the mouse density is above the plague level") whereas the timing of annual control only affects the plague size. Moreover, the authors demonstrate that alternative strategies with similar cost such
as a biannual-seedfall driven control can have similar or better results.

According to me, the manuscript is very clear, well-structured and scientifically
sound. I only have minor comments.

Experimental design

Overall the methods are very well explained, the authors did a great effort to
detail the model as well as its underlying assumptions. Below are a couple
of suggestions:

- Equation 1: I think B should be B(t) as it is time-dependent.
- l. 150: I would explicit the total mortality rate.
- l. 154: in 2015, the authors explored 4 functional responses, I suggest they
mention that based on their previous study they chose Holling II.
- regarding the bait, I found it hard to picture how the unit used relate to the unit
for mice
- l. 192:
- I would mention that it is assume to follow a Normal distribution
- why using a standard deviation of 1 instead of the standard deviation of the time series?
- l.194 I would write "Orongorongo Valley (hereafter OV)"
- l. 206: I would stress out that plague here does not have a specific biological
meaning (unless I'm wrong), it is rather a convenient reference point to compare scenarios.

Validity of the findings

To me, findings are valid and unambiguous. I have two comments:

- Table 2, caption "[...] the minimum expected plague time and plague years
across the different regimes are shown in bold text." Why not doing so for the plague size?
- Figure 1, bottom panel: tick labels are missing.

Additional comments

I would change a few sentences in the introduction and in the abstract to better introduce the example that inspires the model. If I am right, the model is inspired by the house mice dynamics in New Zealand but is thought as a general one. I think this is fair, I would however make this slightly more explicit as I far as I understand, the model has only been validated on this case study.

·

Basic reporting

This manuscript is tightly written and quite clear in outlining modifications to the underlying seasonal mast-driven mouse model first described in Holland et al (2015), with one exception: the double set of inequalities shown in eq 4 seem impossible to satisfy. Perhaps this reflects a typo? The extensive literature cited section is appropriate and well-linked to the text. Figures are clear and easy to follow, as are the tabular data. The results are well-linked to objectives stated in the introduction.

Experimental design

This ms reports on applied population control extension of a previously published model of stochastic seasonal mouse dynamics in New Zealand beech forests. Objectives were to identify the most effective combination of timing and minimal (yet effective) application of pesticide to prevent house mouse outbreaks and their consequent damage to beech forests. Rodent control is an applied ecological problem of considerable concern, but is rarely based on bottom-up driven models of convincing robustness, so the modeling work fills an important gap in the literature and will no doubt be widely cited. The modelling work is based on straightforward modification of a plausible and well-validated consumer-resource model published 3 years earlier. While code was unfortunately not included, my judgement is that it would be simple to replicate the model.

Validity of the findings

The conclusions seem to be well-justified, given the well-validated empirical model framework. My only editorial suggestion is that it would be helpful to clarify why any population density above the median of 2/100 trap nights is defined as a "plague" instead of simply being above the long-term average. It also wasn't clear why the authors restrict their attention to applying pesticide just once a year or once every 2 years in a single massive dose rather than in multiple smaller doses triggered by biomonitoring of either beech mast or mice. In other words: why just these control measures?

Additional comments

I congratulate the authors on a well-formulated and useful modelling paper. I'm sure it will be well-received.

---

## Round 0.2 · accepted · Accept

The authors have carefully considered the suggestions provided and clearly outlined their changes. The manuscript is now suitable for publication.

#